# A Bayesian method to estimate variant-induced disease penetrance

**Brett M. Kroncke**[1,2,3]*, **Derek K. Smith**[4], **Yi Zuo**[4], **Andrew M. Glazer**[1,2], **Dan M. Roden**[1,2,3,5], **Jeffrey D. Blume**[4]

**1** Department of Medicine Vanderbilt University Medical Center, Nashville, Tennessee, United States of America, **2** Vanderbilt Center for Arrhythmia Research and Therapeutics, Vanderbilt University Medical Center, Nashville, Tennessee, United States of America, **3** Department of Pharmacology Vanderbilt University, Nashville, Tennessee, United States of America, **4** Department of Biostatistics Vanderbilt University, Nashville, Tennessee, United States of America, **5** Department of Biomedical Informatics Vanderbilt University Medical Center, Nashville, Tennessee, United States of America

* brett.m.kroncke.1@vumc.org

**Data Availability Statement:** All data and processing scripts are available on the Kroncke lab github website: https://github.com/kroncke-lab/Bayes_BrS1_Penetrance. Additionally the data used are available as a supplement to this

## Abstract

A major challenge emerging in genomic medicine is how to assess best disease risk from rare or novel variants found in disease-related genes. The expanding volume of data generated by very large phenotyping efforts coupled to DNA sequence data presents an opportunity to reinterpret genetic liability of disease risk. Here we propose a framework to estimate the probability of disease given the presence of a genetic variant conditioned on features of that variant. We refer to this as the penetrance, the fraction of all variant heterozygotes that will present with disease. We demonstrate this methodology using a well-established disease-gene pair, the cardiac sodium channel gene *SCN5A* and the heart arrhythmia Brugada syndrome. From a review of 756 publications, we developed a pattern mixture algorithm, based on a Bayesian Beta-Binomial model, to generate *SCN5A* penetrance probabilities for the Brugada syndrome conditioned on variant-specific attributes. These probabilities are determined from variant-specific features (e.g. function, structural context, and sequence conservation) and from observations of affected and unaffected heterozygotes. Variant functional perturbation and structural context prove most predictive of Brugada syndrome penetrance.

## Author summary

The clinical implications for genetic variants, even definitively pathogenic variants, can vary strikingly across individuals. Lack of evidence to estimate the probability of disease from identified genetic variants, especially rare variants, presents a major barrier to integrating genotype information into clinical care. Here we advance an approach to estimate the penetrance, or positive predictive value of the discovery of a genetic variant, in service of advancing the use of genetic information in personalized medicine.

manuscript and through our SCN5A variant browser website: https://oates.app.vumc.org/vancart/SCN5A/.

**Funding:** This research was funded by National Institutes of Health awards R00HL135442 (BMK), P50GM115305 (DMR), K99 HG010904 (AMG), R01HL149826 (DMR), and F32HL137385 (AMG) (https://www.nih.gov/). The funders had no role in study design, data collection and analysis, decision to publish, or preparation of the manuscript.

**Competing interests:** The authors have declared that no competing interests exist.

## Introduction

A major barrier to integrating genotype information into clinical care is accurately linking genetic variants to disease risk. As cheap whole genome, exome, and gene panel sequencing becomes more widely used, the genetics community frequently observes novel, ultra-rare variants—ones carried by a single or few (often related) individuals. Indeed, *most* variants found in large population genome sequencing efforts are novel or ultra rare [1–4]. The number of possible single nucleotide variants in the human genome is in the billions; the number of variants becomes uncountable if insertion and/or deletions (indels) are included. The majority of these discovered variants will never be observed in a sufficient number of heterozygotes to ascertain a causal link with disease. In addition to finding rare variants, large-scale genetic sequencing efforts taking place around the world are identifying greater numbers of individuals, ostensibly unaffected, who carry variants previously thought to be disease-inducing [5, 6]. As a consequence of insufficient heterozygote counts and conflicting annotations, many diagnostic laboratories annotate such variants as "Variants of Uncertain Significance" (VUS), despite more confident past assessments of "Likely Pathogenic" or "Pathogenic" [7–10].

To help assess the impact of genetic variants, the American College of Medical Genetics and Genomics (ACMG) suggests integrating multiple sources of information including population, functional, computational, and segregation data to classify variants [11, 12]. This is consistent with a continuous, Bayesian framework where each additional satisfied classification criterion modifies the probability a variant is causative for disease (pathogenic) or not (benign) [12]. Given the resulting probabilities, a final classification can be made into one of the five categories commonly used to distinguish variants—benign, likely benign, variant of uncertain significance, likely pathogenic, or pathogenic. However, a remaining challenge even after classification is that the clinical implications for definitively pathogenic variants can vary strikingly across individuals, including variable expressivity and incomplete penetrance [13]. We attempt here to address one aspect of this clinical variability by developing a method to estimate variant-induced disease risk.

In this study, we sought to develop a method to estimate the probability of disease given variant-specific information–which we refer to as the penetrance of a variant–and we also provide the uncertainty for that estimate. The pathogenicity of a variant for a specific individual at a given point in time is binary but unknown. This pathogenicity may have a time dependence such as for diseases which present later in life. Penetrance is one metric that captures the degree to which the pathogenicity will manifest as a human phenotype such as a disease or a trait. We provide posterior probability estimates of the penetrance, asymptotic with respect to age, which can be thought of as the positive predictive value of disease given the known variant information. We also provide a 95% credible interval that represents the uncertainty in that estimate. Our method relies on "borrowing strength" or sharing information across variants to produce variant-specific, quantitative penetrance estimates even in the absence of a large number of heterozygotes. These estimates can be especially informative for interpreting rare and novel variants.

We illustrate our approach using the rare cardiac arrhythmia disorder Brugada Syndrome (BrS1 [MIM: 601144]), which is linked to rare loss-of-function variants in the cardiac sodium channel *SCN5* [14]. These variants most commonly act by altering peak sodium current, a parameter of sodium channel function that is readily assessed using *in vitro* methods. By quantitatively integrating multiple features, including *in vitro* functional experiments, information about the three-dimensional protein structure, and previously published variant-classifiers, such as PolyPhen-2 and PROVEAN, we estimate the BrS1 penetrance attributable to individual *SCN5A* variants. The resulting priors, imputed from these

predictive features, can be readily interpreted as hypothetical observations of unaffected and affected heterozygotes.

## Results/Discussion

Variants in *SCN5A* have been associated with BrS1 since 1998,[15] some variants affecting almost all known heterozygous individuals, some variants conferring only modestly increased risk, and others have no influence on arrhythmia presentation [14, 16, 17]. *SCN5A* variants that do not influence the gene in any way do not predispose or protect against BrS1, e.g. many synonymous variants. These variants therefore have a relatively low penetrance of the arrhythmia, similar to the general population. *SCN5A* variants that produce no sodium current result in a higher fraction of heterozygotes presenting with BrS1, much higher than in the general population [18]. However, BrS1 presentation, as for nearly all inherited diseases, is not homogeneous even amongst heterozygotes of *SCN5A* haploinsufficiency alleles. In fact, even highly penetrant variants such as p.Glu1784Lys and p.Glu1784Lys still leave some heterozygotes unaffected: 100% penetrance is extremely rare [18].

Our hypothesis is that variant-specific features (e.g. variant-induced changes in function and location in structure) contain *information equivalent* to clinically phenotyping heterozygotes and can therefore be used to inform the prior distribution in a Bayesian framework. This prior distribution is combined directly with clinically phenotyped heterozygotes (the likelihood function) to produce more accurate estimates of disease risk probability (posterior penetrance; Fig 1) via Bayes theorem. To demonstrate this approach, we developed an expectation maximization approach (EM), detailed in the Materials and Methods section, and applied it to a previously generated dataset of *SCN5A* features and BrS1 phenotype counts [18] (supplemented with reports published within the last year) to estimate BrS1 penetrance using *SCN5A* variant-specific features. This process yielded a total of 1,439 unique variants with at least 1 observed heterozygote, BrS1 was diagnosable in 857 individuals heterozygous for 387 unique variants (S1–S3 Figs). BrS1 penetrance priors informed by the predictive features listed in S1 Table adjust and narrow the uncertainty, as shown in Fig 1.

### Precision and accuracy of BrS1 penetrance priors

To evaluate performance over the distribution of BrS1 prior penetrances (S5 Fig), we plotted the difference between prior mean and posterior mean BrS1 penetrance as a function of the average between the two estimates (Fig 2). The resulting Bland-Altman difference plot seen in Fig 2 indicates scatter evenly distributed with under and over predicted BrS1 penetrance as a function of prior mean penetrance. This suggests the predictive priors are reasonably calibrated and have no systematic biases in the range of BrS1 mean penetrance estimated. We additionally compared linear regression models trained on a limited subset of features/covariates with the BrS1 mean posterior, $\frac{\text{BrS1 cases} + \alpha_{\text{prior}}}{\text{total heterozygotes} + \alpha_{\text{prior}} + \beta_{\text{prior}}}$ (where $\alpha_{\text{prior}}$ and $\beta_{\text{prior}}$ are the tuning parameters for the beta-binomial distribution and are set equivalent to the number of affected and unaffected individual heterozygotes in the prior), as the dependent variable; both empirical and EM priors were evaluated as indicated in Table 1. Peak current and penetrance density (a modification of a structure-derived feature we developed previously[19]; see S1 Text) contain orthogonal information as can been seen by the differences in coefficient of determination, $R^2$, for models built using each or both predictors (Table 1). The relatively small improvement in $R^2$ when all predictors are included suggests most information contained in the sequence-based predictive features is recapitulated by both peak current and penetrance density.

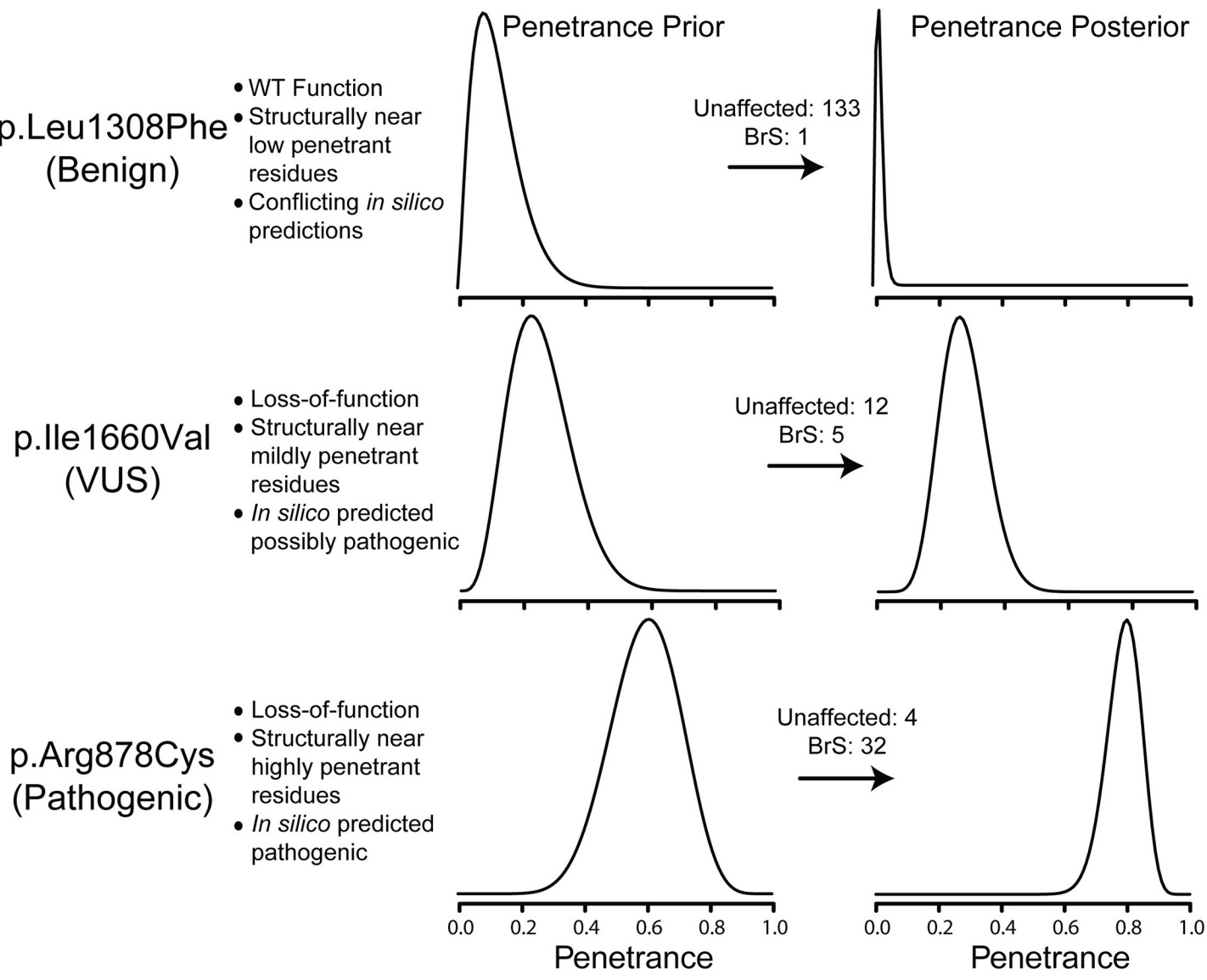

**Fig 1. Penetrance priors are informed by variant-specific features.** Probability density (y-axis) versus penetrance (x-axis) for three selected *SCN5A* variants where peak current, penetrance density, and *in silico* classification are known. Numbers of affected and unaffected individuals reported are presented for each variant. Penetrance priors are low for c.3922C>T (p.Leu1308Phe; Benign according to ClinVar), moderate for c.4978A>G (p.Ile1660Val; VUS), and higher for c.2632C>T (p. Arg878Cys; Pathogenic). When variant-specific data are known, the penetrance estimate is adjusted to reflect the penetrance probability consistent with variants with similar features.

## Inclusion of individuals from gnomAD

Individuals in gnomAD are mostly unaffected, given the rarity of BrS; however, the data available from that resource could be contaminated with individuals presenting with BrS, though likely at or near the rate in the general public. To test the sensitivity of our results to this type of misclassification, we randomly switched individuals from unaffected (gnomAD) to BrS cases for each variant and examined the change in penetrance due to misclassification. We did this with 24 and 240 misclassified cases. With 24 misclassifications, the median rate of penetrance change is 0.4% and the expected number of variants with a penetrance change is 6. The average mean absolute difference in penetrance change is 0.02% (first quartile of 0.0014% and

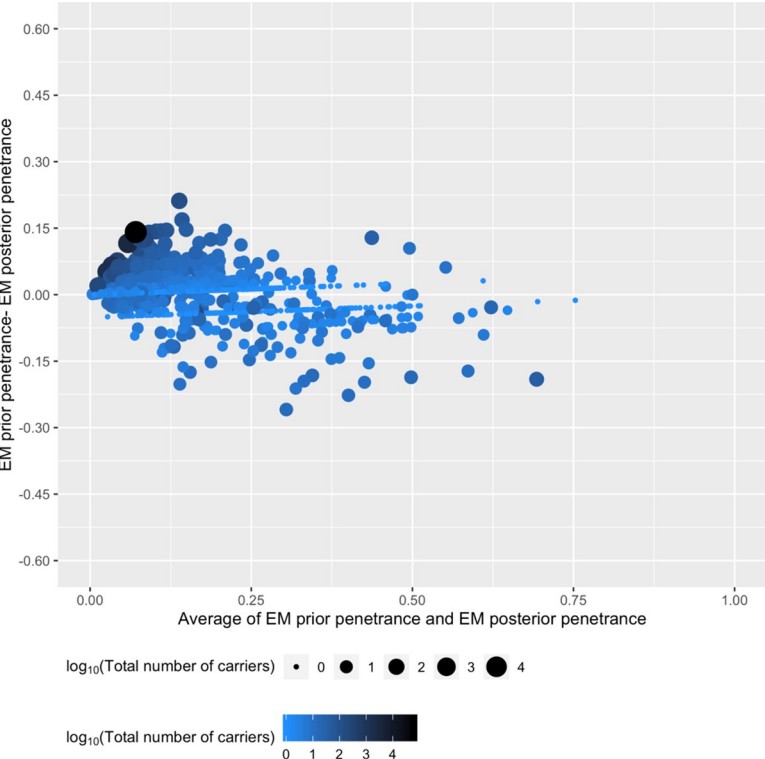

**Fig 2. Bland-Altman plot between EM prior and EM posterior mean penetrances for all *SCN5A* variants.** To assess the performance of the EM prior, we used a Bland-Altman plot to compare the mean BrS1 penetrance estimated from the EM prior and from the EM posterior, the y-axis is the difference between the two and the x-axis is the average between the two. For each plotted point, both color and radius indicate the $\log_{10}$ of the total number of heterozygotes present in the dataset. The relatively consistent scatter about y = 0 suggests no systematic biases present in the EM prior mean BrS1 estimates.

third quartile of 0.02%). With 240 misclassifications, the median rate of penetrance change is 2%, and the expected number of variants with a penetrance change is 28. The average mean absolute difference in penetrance change is 0.2% (first quartile of 0.1% and third quartile of 0.3%). These results suggest minimal influence of small or modest misclassification rates on penetrance estimates.

**Table 1. Weighted $R^2$ from EM prior means to Empirical/EM posterior means.** Models trained with displayed subsets of features using the same subset of variants, where covariates listed in S1 Table are known.

| Features | Empirical† | EM† |
|---|---|---|
| Peak Current | 0.22 [0.12–0.34; 155] | 0.35 [0.24–0.45; 20] |
| Penetrance Density | 0.35 [0.20–0.49; 113] | 0.66 [0.53–0.76; -124] |
| Peak Current and Penetrance Density | 0.43 [0.27–0.57; 88] | 0.76 [0.66–0.83; -201] |
| All Features | 0.44 [0.28–0.59; 90] | 0.78 [0.69–0.85; -218] |
| Sequence-based Features | 0.12 [0.06–0.19; 189] | 0.20 [0.12–0.28; 74] |

†Weighted $R^2$ [95% Confidence Interval; Akaike information criterion], weighted by inverse beta-binomial variance capped at the 9th decile as described in the methods section

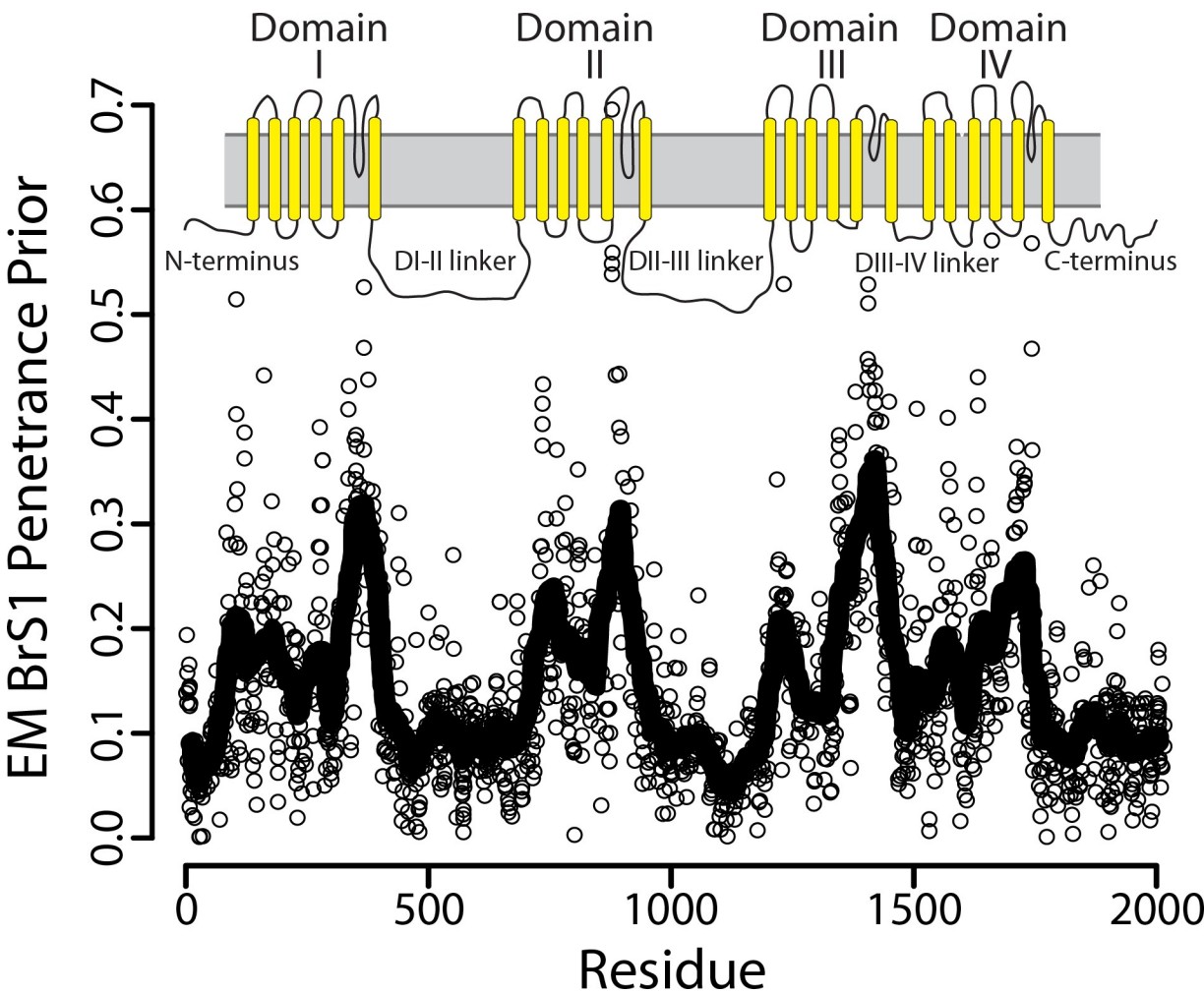

**Fig 3. Prior mean BrS1 penetrance reflects the protein topology of Na$_V$1.5.** The predicted mean BrS1 penetrance from the converged expectation maximization (EM) algorithm. The line across the plot is a predicted mean BrS1 penetrance averaged over 30 neighboring variants. Topology diagram is shown above with transmembrane helices indicated by yellow lines and membrane indicated as a grey rectangle. Note the four largest, distinct peaks correspond to the four structured, transmembrane domains of the channel, with an especially steep peak at the selectivity filter and pore. Though estimated distances in three-dimensional space between residues is used to construct the BrS1 penetrance density, structural data are not explicitly used in the BrS1 penetrance prior and so the recapitulation of the structure is not assured.

## Structure and peak current improve prediction of penetrance

The resulting prior BrS1 mean penetrance estimates reflect the known topology of Na$_V$1.5 (protein product of *SCN5A*; Fig 3), with the sodium channel pore and selectivity filter inducing a greater disease burden as previously observed [18, 20]. Fig 4 examines in greater detail a small region within domain III (D-III), showing the 95% credible interval of BrS1 penetrance both before (prior) and after (posterior) adding heterozygote counts listed on the left. The selectivity filter has the highest average BrS1 prior and posterior, also true for domains I, II, and IV (Fig 3). Towards the intracellular side of the D-III S6 helix, there are fewer variants with high BrS1 penetrance. This trend can also be seen in S6 Fig which shows an increase in variants associated with BrS1 that depends on membrane depth of the variant. These results support our assertion that variant-specific predictive features of variant-induced functional perturbation and structural context contain information equivalent to

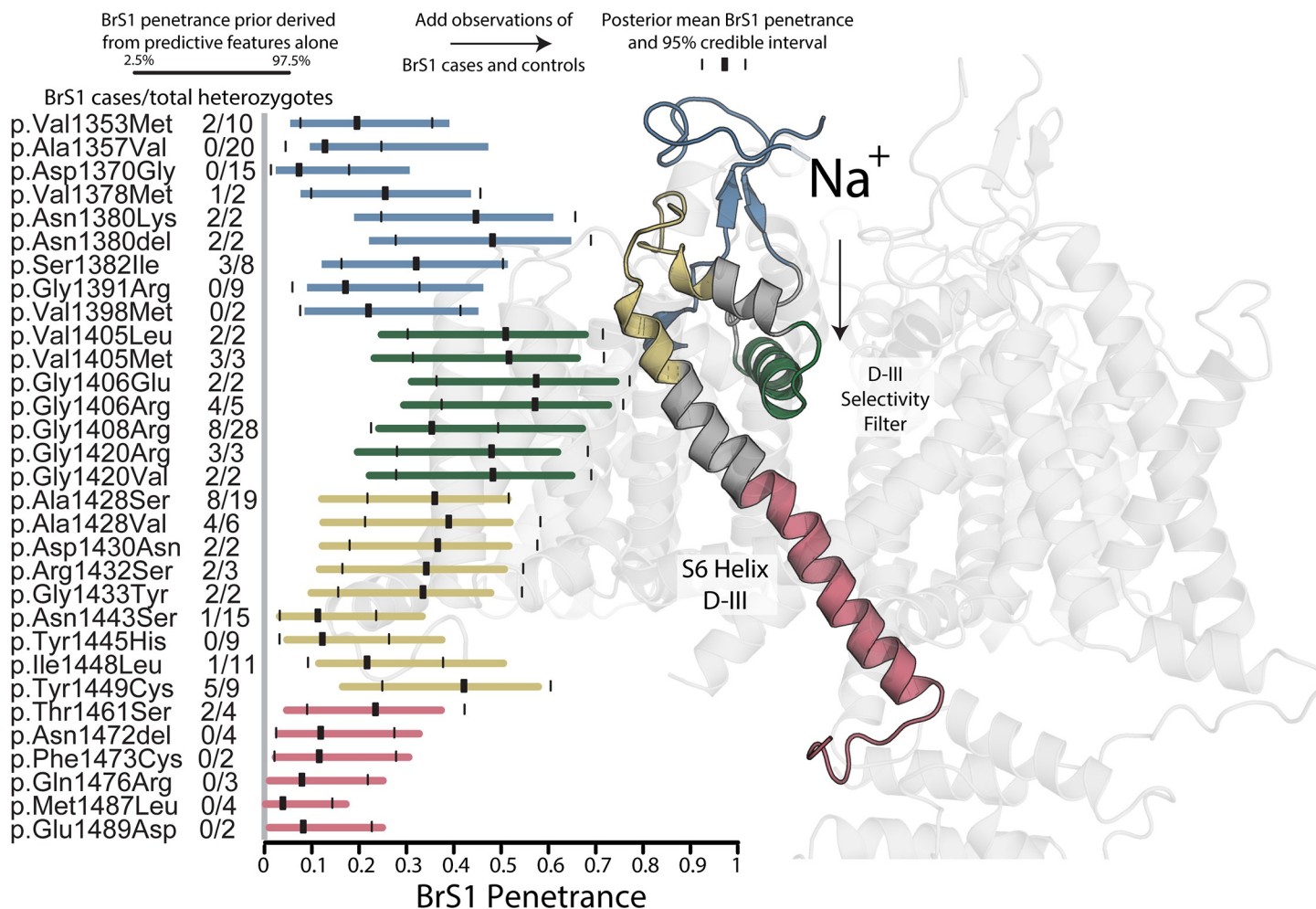

**Fig 4. Sample of BrS1 penetrance prior 95% credible intervals.** Left: *SCN5A* variants with more than one heterozygote in our dataset are plotted with prior 95% credible intervals (colored bars) and mean posteriors (black rectangles) with posterior 95% credible intervals (black lines). Right: a model of the *SCN5A* protein product, Na$_V$1.5, is shown with the regions highlighted in blue, green, gold, and red, corresponding to the colors of the variant prior 95% credible intervals shown to the left, which are analogous to the penetrance probability distributions shown on the y-axes in Fig 1. Variants near the D-III pore selectivity filter have a much higher prior and posterior BrS1 penetrance compared to residues near the D-III/D-IV linker. This is expected since the selectivity filter pore helices contain the most compacted region of the protein and also are responsible for the ion conduction and are therefore most sensitive to substitution. In fact, the highest density of variants with non-zero BrS1 penetrance lie at this depth in the membrane (S6 Fig). Variants listed are c.4057G>A (p.Val1353Met), c.4070C>T (p.Ala1357Val), c.4109A>G (p.Asp1370Gly), c.4132G>A (p.Val1378Met), c.4140C>G (p.Asn1380Lys), c.4137_4139CAA (p.Asn1380del), c.4145G>T (p.Ser1382Ile), c.4171G>A (p.Gly1391Arg), c.4192G>A (p. Val1398Met), c.4213G>C (p.Val1405Leu), c.4213G>A (p.Val1405Met), c.4217G>A (p.Gly1406Glu), c.4216G>A (p.Gly1406Arg), c.4222G>A (p.Gly1408Arg), c.4258G>C (p.Gly1420Arg), c.4259G>T (p.Gly1420Val), c.4282G>T (p.Ala1428Ser), c.4283C>T (p.Ala1428Val), c.4288G>A (p.Asp1430Asn), c.4296G>T (p. Arg1432Ser), c.4297G>T (p.Gly1433Trp), c.4328A>G (p.Asn1443Ser), c.4333T>C (p.Tyr1445His), c.4342A>C (p.Ile1448Leu), c.4346A>G (p.Tyr1449Cys), c.4381A>T (p.Thr1461Ser), c.4414_4417AAC (p.Asn1472del), c.4418T>G (p.Phe1473Cys), c.4427A>G (p.Gln1476Arg), c.4459A>C (p.Met1487Leu), c.4467G>T (p. Glu1489Asp).

clinically phenotyping individuals heterozygous for these variants. The interchangeability of this information was additionally demonstrated recently by taking the reverse approach: functionally characterizing variants with different estimates of BrS1 penetrance [21]. In these experiments, Glazer et al. found that variants with a higher estimated BrS1 penetrance had a higher probability of producing a variant-induced loss-of-function protein phenotype (Fig 3A in reference [21]).

## A modified Bayesian approach to estimate BrS penetrance

A typical Empirical Bayes approach combines information across all variants to estimate a *single* prior distribution and estimate a variant-specific posterior penetrance from that prior. These estimates assume all variant effects have the same prior and therefore shrink towards a global mean across all variants. Here we put forward a method to model the penetrance for *each* variant using variant-specific predictive features. The resulting penetrance and uncertainty estimates yield a posterior that can be re-used as variant-specific prior (interpretable as equivalent to hypothetical observations of affected and unaffected heterozygotes) in a classical Bayesian updating scheme. This information is accessible before clinically phenotyping a single heterozygote; example estimates of high BrS1 penetrance [c.4213G>C (p.Val1405Leu), c.4259G>T (p.Gly1420Val), and c.4258G>C (p.Gly1420Arg)] and low BrS1 penetrance [c.4418T>G (p.Phe1473Cys), c.4459A>C (p.Met1487Leu), and c.4467G>T (p.Glu1489Asp)] are seen in Fig 4.

## Comparison between penetrance prediction and ACMG variant classification

We put forward a method to estimate the probability that an *SCN5A* variant will manifest in BrS1 for a given patient (our 'risk score'), and uncertainty for that score, conditioned on variant attributes. We are not assessing the causality of the variant and its attributes on the manifestation of disease, but rather their association. Hence, our framework diverges from that of the ACMG, quantitated by Tavtigian et al. 2018. For example, in our formulation, a VUS with many affected heterozygotes would have the same probability distribution as a pathogenic variant with many affected heterozygotes [provided the number of observations of cases and controls is the same and the other predictive covariates (variant attributes) are the same]. If there are comparatively few heterozygotes of the VUS, given the same predictive covariates, greater uncertainty would be reflected by a wider distribution of penetrance probability (Fig 1). In addition, our calculation is agnostic to origin, *de novo* or inherited, and therefore does not consider this evidence (though this information may additionally inform an estimate of penetrance and therefore warrants further investigation). We also do not treat null variants here. For our purposes of building variant-specific, data-driven penetrance priors, null variants have relatively little variance in the predictive covariates and therefore contribute less to our analysis. In future work we will additionally attempt to include these features.

## Prospects for applications of this method

Our approach provides a risk score for disease, in this case, for BrS1. However, Brugada syndrome has degrees of electrophysiologic phenotypes and symptoms. We envision being able to predict these degrees of clinical phenotype from variant-specific properties in the future by integrating electronic health records with linked genetic data. However, at present, these granular electrophysiologic and symptom data are not available for a number of unique heterozygotes and unique variants sufficient for statistical analysis. Beyond *SCN5A* and BrS1, a reasonable next step would involve the 59 genes for which the ACMG recommends clinical diagnostic laboratories report secondary variant discovery. Of these, 36 have greater than or equal to 20 missense "pathogenic"/"likely pathogenic" variants in ClinVar,[22] suggesting that many variants are described in the literature and can be curated in a similar manner to *SCN5A*. It is also important to note that the penetrance estimates derived in our approach are not static and will continue to be refined as additional data become available, i.e. phenotype

data from case reports and large biobank projects, additional *in vitro* functional studies, and improved computational and structural predictors [13, 23–26].

## Limitations

Our approach provides a risk score for disease, in this case BrS1, analogous to a diagnostic test (might patient X develop BrS1 given they have variant Y). If we know patient X already has BrS1, we can use their data to inform other individuals' risk scores, but we cannot use our approach to absolutely determine the role of variant Y manifesting disease. One application of our approach is that we can examine the ratio P(BrS1|SCN5A Variant X)/P(BrS1|wild-type SCN5A) to see if the data better support that variant X is on the causal pathway to disease. But we caution that this approach is imperfect; it does not allow for variants to interact, for example. Additionally, while clinical evidence affirms a strong relationship between *SCN5A* variants and BrS1, many genetic and environmental factors influence the ultimate presentation of BrS1 in an individual [13, 27, 28]. Not accounting for additional demographic, genetic, or environmental factors certainly increased the noise in our analysis. To counter this as best as possible, we included the maximum number of carriers for the maximum number of unique variants. Finally, we recognize the likely bias intrinsic to compiling a list of affected and unaffected heterozygotes in the manner outlined in the methods section above; however, the most probable manifestation of these biases would be the loss of an observable relationship between the predictive features and penetrance, not the creation of a spurious relationship.

## Conclusions

We advance a method to estimate a degree of clinical heterogeneity in variant impact, incomplete penetrance. Here we have demonstrated how BrS1 penetrance can be estimated with high accuracy and precision. Using a Bayesian framework to estimate penetrance allows us to quantitatively integrate clinical phenotypic data with variant-specific functional measurements, variant classifiers, and sequence- and structure-based features to accurately estimate penetrance. This method can be extended to other genes and disorders in order to enable quantitative interpretation of variants probabilistically and quantitatively [24, 29].

## Materials and methods

These analyses focus on the *SCN5A* gene, where individual variants are known to influence the clinical presentation of the autosomal dominant arrhythmia Brugada Syndrome (BrS1) [16, 17]. We define cases as individuals with either a spontaneous or drug-induced ECG BrS1 pattern, ST-segment abnormalities, as reported in each publication [18, 30]. Penetrance is defined as the fraction of individuals who carry a variant that also present with a disease. This can be extracted from literature reports when multiple variant heterozygotes have been reported. We do not observe the actual penetrance for any given variant; however, we can estimate BrS1 penetrance for each variant as the average posterior penetrance denoted as the following:

$$Mean\ Posterior\ Penetrance = \frac{\alpha + \alpha_{prior}}{\alpha + \beta + \alpha_{prior} + \beta_{prior}} \qquad \text{Eq 1}$$

Where $\alpha$ is the number of variant heterozygotes diagnosed with BrS1 (or BrS1 cases) and $\beta$ is the number of unaffected heterozygotes of the same variant (or controls). As the total number of observed heterozygotes increases, the estimated penetrance converges to the traditional

definition. The mean posterior penetrance can be thought of as a shrunken estimate of the observed penetrance [31], especially for variants with small numbers of known heterozygotes.

To generate priors from our available data, we use a variation of the expectation maximization (EM) algorithm [32]. Our modified EM algorithm is an iterative technique composed of three steps: 1) calculate the expected penetrance from an empirical Bayes penetrance model, 2) fit a regression model of our estimated penetrance on variant-specific characteristics by maximum likelihood (Eq 2, below) and 3) revise our estimate of the BrS1 penetrance prior using the fit from step 2 then iterate steps 2–3 until convergence criteria are satisfied (S7 Fig).

$$
\begin{aligned}
\text{Penetrance Estimate}_i \\
= \beta_0 + \beta_1 (\text{Peak Current})_i + \beta_2 (\text{Penetrance Density})_i \\
+ \sum_n \beta_{i,n} (\textit{In Silico Variant Classifiers})_{i,n} + \varepsilon_i
\end{aligned}
\qquad \text{Eq 2}
$$

Here peak current is an *in vitro* measurement of the maximum current through a channel (normalized to wild type), penetrance density is a structure-based metric [19] detailed in the S1 Text, and *in silico* variant-classifiers is a vector populated with commonly used variant classification servers such as PROVEAN and PolyPhen (see below); all predictors used are continuous, not categorical or binary (S1 Table). The fitted model is then used to generate an updated prior distribution and, by addition of observed cases and controls for each variant, a subsequent posterior expected penetrance. The updated posterior penetrance is then used to build a new fitted model and further refine the posterior expected penetrance. This procedure is iterated until it converges to the maximum likelihood solution (S7 Fig). Using a beta-binomial model to estimate penetrance, the prior parameters ($\alpha_{\text{prior, EM}}$ and $\beta_{\text{prior, EM}}$, both functions of the features listed in S1 Table) are identifiable from a predicted penetrance point estimate and its associated variance. For comparison, we generated predicted penetrance values using a standard empirical Bayes method which generated a single empirical prior for all variants, $\alpha_{\text{prior, empirical}}$ and $\beta_{\text{prior, empirical}}$ equal to 0.45 and 2.73, respectively (called empirical prior throughout the text, S8 Fig). To test our predictions, we compare our EM penetrance priors, $\frac{\alpha_{\text{prior,EM}}}{\alpha_{\text{prior,EM}} + \beta_{\text{prior,EM}}}$, to the posterior mean penetrance derived by adding BrS1 cases and controls for each variant to the empirical prior, $\frac{\textit{BrS1 cases} + \alpha_{\text{prior,empirical}}}{\text{Total heterozygotes} + \alpha_{\text{prior,empirical}} + \beta_{\text{prior,empirical}}}$, or the EM prior, $\frac{\textit{BrS1 cases} + \alpha_{\text{prior,EM}}}{\text{Total heterozygotes} + \alpha_{\text{prior,EM}} + \beta_{\text{prior,EM}}}$.

## Collection of the SCN5A variant dataset

The dataset was curated from 711 papers in a previous publication [18], to which we added an additional 45 papers on *SCN5A* that had been published since the previous dataset was constructed. Briefly, we searched publications for the number of heterozygotes of each variant mentioned, the number of unaffected and affected individuals with diagnosed BrS1, and variant-induced changes in channel function, if reported; all recorded values of channel function were normalized to wild-type values reported in the same publications. We supplemented this dataset with all *SCN5A* variants in the gnomAD database of population variation (http://gnomad.broadinstitute.org/; release 2.0) [33]. Due to the rarity of BrS1 (~1 in 10,000) [34], all heterozygotes found in gnomAD were counted as unaffected. An interactive version of the dataset, the *SCN5A* Variant Browser, is available at https://oates.app.vumc.org/vancart/SCN5A/. We further collected *in silico* pathogenicity predictions from three commonly used servers: SIFT [35], Polyphen-2 [36], and PROVEAN [37]. We also include basic local alignment search tool position-specific scoring matrix (BLAST-PSSM)[38] for *SCN5A* and the per residue evolutionary rate [39], previously shown to have predictive value for predicting

functional perturbation for the cardiac potassium channel gene *KCNQ1* [40], and point accepted mutation score (PAM) [41]. Additionally, we leveraged structures of the *SCN5A* protein product and derived a penetrance density as previously described (see S1 Text for details) [19]. In-frame indels are treated as missense variants. We include these variants as variations at a residue where the indel starts, and only note whether they are an insertion or deletion. Some of these variants have functional data available and their penetrance densities are calculated from the residue starting the indel. These are simplifications to enable an analysis of as many variants and heterozygote individuals as possible. For these variants, we did not include *in silico* pathogenicity predictions. We included compound heterozygotes (individuals with more than one *SCN5A* variant) as separate records when these data are available, though these were very rare. Additionally, our inclusion criteria are not modified by relatedness. We did not include intronic variants in our analysis. The dataset is available in S2 Table.

## Initial Empirical Bayes beta-binomial prior penetrance calculation

Using the data from the aforementioned literature curation [18], we estimated the penetrance for each observed variant using a beta-binomial empirical Bayes model. To calculate the empirical BrS1 penetrance prior, we calculated $\alpha_{prior, empirical}$ and $\beta_{prior, empirical}$ by finding the weighted mean penetrance over all variants in the dataset and estimating the variance. Weighting was done using the following equation:

$$w = 1 - \frac{1}{0.01 + number\ of\ heterozygotes} \qquad \text{Eq 3}$$

Eq 3 ensures variants with a greater number of total heterozygotes (and therefore higher confidence in penetrance estimate) had a greater weight in the preliminary analysis. We then estimated the variance in penetrance as the mean squared error (MSE) between the estimated penetrance mean and the observed penetrance from Eq 1 with $\alpha_{prior}$ and $\beta_{prior}$ equal to zero. With these estimated mean and MSE-derived variance, the empirical prior penetrance was calculated to be an $\alpha_{prior}$ and $\beta_{prior}$ equal to 0.45 and 2.73, respectively. The variant-specific empirical posterior for each variant was then calculated by adding observed heterozygote counts of affected (BrS1 cases) and unaffected to $\alpha_{prior, empirical}$ and $\beta_{prior, empirical}$, respectively, and the resulting posterior mean penetrance was used as the dependent variable of the subsequent regression model (Eq 2). The inverse variance of the estimated posterior beta distributions capped at the ninth decile determined in this step were used to weight subsequent regression models and Pearson $R^2$ calculations.

## Expectation maximization Bayesian beta-binomial penetrance predictions

To deal with missing data in a prediction model, we followed the approach outlined in Mercaldo and Blume [42] which avoids multiple imputation but guarantees maximum predictive accuracy across missing data patterns. In short, for every missing data pattern, we estimate a separate prediction model. For example, p.His558Arg, where penetrance density, in silico predictors, and functional data are all available, the estimate of penetrance is regressed on all other variants where all of these covariates are available (n = 238). For p.Try1449Cys, however, only penetrance density and *in silico* predictors are available, so only those covariates are used in the regression (n = 1,382; much higher since functional data have been collected for relatively few variants). The models were built with a linear regression pattern-mixture algorithm, updating posterior mean penetrances iteratively until the resulting estimated mean penetrance, $\mu = \frac{\alpha_{prior,EM}}{\alpha_{prior,EM} + \beta_{prior,EM}}$, changed by < 0.01% from the previous iteration. This process

typically converged within eight iterations. For variant, i, the variance was estimated from this converged EM mean penetrance according to (Eq 4):

$$\sigma_i = \frac{\mu_i(1 - \mu_i)}{1 + v} \qquad \text{Eq 4}$$

We then adjusted v, equivalent to the number hypothetical observations of clinically phenotyped heterozygotes, to balance overcoverage of variants with low to moderate BrS1 penetrance and poorer coverage of variants with high estimated mean penetrance, resulting in a range of v, from approximately 15 to 20 (see S2 Text for details; S9–S12 Figs). All analyses were performed using the datasets provided in S2 Table and at the Kroncke lab GitHub site: https://github.com/kroncke-lab/Bayes_BrS1_Penetrance.

## Supporting information

**S1 Text. Detailed explanation of penetrance density calculation.**
(DOCX)

**S2 Text. Detailed explanation of how 'v' from Eq 4 was determined.**
(DOCX)

**S1 Table. *SCN5A* variant-specific features used to predict BrS1 penetrance.**
(DOCX)

**S2 Table. *SCN5A* dataset.** All data used to estimate BrS1 penetrance including covariates are included in the accompanying dataset.
(CSV)

**S1 Fig. Histogram of the frequency of variants (y-axis) with different number of individuals diagnosed with Brugada syndrome (x-axis).** Most variants have only a single heterozygote diagnosed with BrS; however, there are over 10 variants with 10 or more heterozygotes diagnosed with BrS.
(PNG)

**S2 Fig. Frequency of variants (y-axis) with different counts in gnomAD (x-axis).** The x-axis is truncated at 350. There are 10 variants with greater than 350 carriers.
(PNG)

**S3 Fig. Frequency of variants (y-axis) with different observed BrS penetrances (x-axis).** Most variants have either exactly 0 or exactly 1 observed BrS penetrance, at odds with both the known background rate of BrS in the general public (approximately 1 in 10,000–20,000) and with the extreme rarity of any variant having 100% penetrance.
(PNG)

**S4 Fig. Bland-Altman plot between EM posterior mean BrS penetrances and observed BrS penetrance for SCN5A variants with at least 15 heterozygotes.** The relatively narrow spread along the y-axis suggests reasonable agreement between the two estimates of BrS penetrance. With the cutoff of at least 15 heterozygotes, there are relatively few variants with an expected penetrance of greater than 10%.
(PNG)

**S5 Fig. Histogram of BrS1 penetrance imputed EM prior means and associated upper and lower bounds to 95% credible interval from pattern mixture models.** Plotted are BrS1 mean penetrances from imputed EM priors ("Predicted", green) and upper (red) and lower (blue)

bounds to associated 95% credible intervals from those imputed EM priors.
(PNG)

**S6 Fig. *SCN5A* pathogenic and benign variants cluster in space.** Rate of variants with high BrS1 penetrance (>20%, blue) or low BrS1 penetrance (<10%, red) in a model of the SCN5A protein product. Each bar represents a histogram of variants associated with each disease within a 5Å slice within the membrane (divided by the total number of residues within the slice), boxes at each of the four corners represent residues not modeled (only 33 residues were not modeled in the extracellular loops). There is a relative paucity of low BrS1 penetrance variants within the structured transmembrane region and the relative abundance of high BrS1 penetrance in the same region. The rate of high BrS1 penetrance variants is higher in the extracellular half of the protein molecule likely due to more compacting of residues in the top half of the pore domain as well as proximity to the ion selective element (selectivity filter). Amino acid substitutions in these regions therefore more often have a disruptive influence.
(PNG)

**S7 Fig. Generation of empirical and EM priors.** The modified EM algorithm is an iterative technique composed of two steps: 1) calculate the expected penetrance from an empirical Bayes penetrance model and 2) fit regression of our estimated penetrance on variant-specific characteristics by maximum likelihood. The fitted model is then used to generate an updated, imputed prior and subsequent posterior expected penetrance and this process is iterated until it converges to the maximum likelihood solution, when the new mean penetrance changed by less than 1% from the previous iteration. The variance is then estimated according to Eq 4 as explained above.
(PNG)

**S8 Fig. BrS1 penetrance probability versus penetrance for the empirical prior.**
(PNG)

**S9 Fig. Estimated coverage rates for each *SCN5A* variant versus sampled true penetrance.** Coverage rate was calculated as defined above. Color and radius indicate the $\log_{10}$ of the total number of heterozygotes present in the dataset. The tuning parameter Eq 4 was set to $v = 7$. There is overcoverage (greater than 95%) for variants with high and low BrS1 penetrance indicating an overestimate of the variance.
(PNG)

**S10 Fig. Estimated coverage rates for each *SCN5A* variant versus sampled true penetrance.** Coverage rate was calculated as defined above. Color and radius indicate the $\log_{10}$ of the total number of heterozygotes present in the dataset. The tuning parameter Eq 4 was set to $v = 14$. There is overcoverage for the majority of variants, though some variants are now outside the 95% credible interval.
(PNG)

**S11 Fig. Estimated coverage rates for each *SCN5A* variant versus sampled true penetrance.** Coverage rate was calculated as defined above. Color and radius indicate the $\log_{10}$ of the total number of heterozygotes present in the dataset. The tuning parameter Eq 4 was set to $v = 19$. Overcoverage is reduced especially for residues with very low or very high BrS1 penetrance, indicating an appropriate estimate of variance.
(PNG)

**S12 Fig. Estimated coverage rates for each *SCN5A* variant versus sampled true penetrance.** Coverage rate was calculated as defined above. Color and radius indicate the $\log_{10}$ of the total

number of heterozygotes present in the dataset. The tuning parameter Eq 4 was set to $v = 99$. Variant undercoverage is much more prevalent and distributed evenly across variants with low to high BrS1 penetrance indicating an overestimate of variance.
(PNG)

## Author Contributions

**Conceptualization:** Brett M. Kroncke, Derek K. Smith.

**Data curation:** Brett M. Kroncke, Andrew M. Glazer.

**Formal analysis:** Brett M. Kroncke, Derek K. Smith, Yi Zuo.

**Funding acquisition:** Brett M. Kroncke.

**Investigation:** Brett M. Kroncke.

**Methodology:** Brett M. Kroncke, Derek K. Smith, Yi Zuo, Jeffrey D. Blume.

**Project administration:** Brett M. Kroncke.

**Resources:** Brett M. Kroncke.

**Software:** Brett M. Kroncke.

**Supervision:** Brett M. Kroncke, Jeffrey D. Blume.

**Validation:** Brett M. Kroncke.

**Visualization:** Brett M. Kroncke, Yi Zuo.

**Writing – original draft:** Brett M. Kroncke.

**Writing – review & editing:** Brett M. Kroncke, Derek K. Smith, Yi Zuo, Andrew M. Glazer, Dan M. Roden, Jeffrey D. Blume.

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
