## [Decision Letter · Decision Letter 0]

31 Dec 2019

Dear Dr Kroncke,

Thank you very much for submitting your Research Article entitled 'A Bayesian method to estimate variant-induced disease penetrance: moving beyond a dichotomous view of variant pathogenicity' to PLOS Genetics. Your manuscript was fully evaluated at the editorial level and by independent peer reviewers. The reviewers appreciated the attention to an important problem, but raised some substantial concerns about the current manuscript. Based on the reviews, we will not be able to accept this version of the manuscript, but we would be willing to review again a much-revised version. We cannot, of course, promise publication at that time.

If you decide to revise the manuscript for further consideration at PLOS Genetics, please aim to resubmit within the next 60 days, unless it will take extra time to address the concerns of the reviewers, in which case we would appreciate an expected resubmission date by email to plosgenetics@plos.org.

[LINK]

We are sorry that we cannot be more positive about your manuscript at this stage. Please do not hesitate to contact us if you have any concerns or questions.

Yours sincerely,

Leslie Biesecker

Guest Editor

PLOS Genetics

Hua Tang

Section Editor: Natural Variation

PLOS Genetics

The paper has been reviewed by two anonymous reviewers and I have added some comments here. The opinions of the peer reviewers was widely disparate, but both identified major issues.  I add some comments below.

1. You appear to be collapsing the pathogenicity of the variant and the penetrance function of a variant into a single concept. It is not clear to me why this is the correct approach. A VUS can have high (apparent) penetrance and a pathogenic variant could have ~50% penetrance. Your construct would appear to evaluate these the same.

2. You are inaccurately characterizing the Richards et al and Tavtigian et al heuristics. The former (implicitly) and the latter (explicitly) yield pathogenicity assertions in a nearly continuous probabilistic fashion, but have chosen to use a post-hoc reporting system of five tiers. Anyone using Richards et al can readily see that a variant with one strong and three supporting criteria for pathogenicity is more likely to be pathogenic than would be a variant with one strong and two supporting pathogenic (likely pathogenic criterion iii). You also inaccurately describe this as a three tier system by collapsing P and LP into one category and B and LB into one category and then criticizing it for being too coarse – “…with variants not confidently placed in either of these categories classified as VUS.” Emphasis added. The Tavitigian et al heuristic showed how Richards et al can be transformed into a (nearly) continuous output posterior probability – they are the same. This characterization is also problematic – “Here we advance an approach that quantitates degree of pathogenicity, probabilistically…” but Tavtigian did that as well and you seem to be suggesting that your work in this respect is novel. Your figure 5 is especially problematic in this regard. Even worse that the (incorrect, in my view) collapsing of P & LP, you simply label them both as P. You should also be aware of a larger issue, which is that there is a strongly held view in medicine that clinicians do not want posterior probabilities at all – they want yes-no answers (you and I don’t accept this view, but I think we are, for the time being, outnumbered). Thus, this is a complicated issue. I think this thrust of your article is not justified and not supported by a review of prior work. Instead, what I think you are doing is describing a novel formulation of a conditional probability and using a different Bayesian formulation than did Tavtigian. That to me seems like a perfectly valid and useful approach and would simplify and clarify your paper.

3. You state that “By quantitatively integrating multiple features, including in vitro functional experiments, information about the three-dimensional protein structure, and previously published variant classifiers, we estimate the BrS1 penetrance attributable to individual SCN5A variants.” This seems to suggest that you are supplanting Richard/Tavtigian PS3 (in vitro functional experiments) PP3 (information about the three-dimensional protein structure) with your measures. I am troubled by ‘previously published variant classifiers’ – I have no idea what that means. Later you say this: “Our hypothesis is that features such as variant-induced changes in function, sequence conservation, and location in structure, contain equivalent information to clinically phenotyping carriers and can therefore be used to calculate a penetrance prior. This seems to suggest three criteria, but different than the first three I quote above. The first again seems to be PS3 and are the second and third PP3?

4. I am also puzzled as to why this is considered a prior probability. To me, it seems like a conditional probability. You seem to have a very different concept of priors and posteriors than does Tavtigian et al – this needs to be much better explained.

5. Your system would not appear to take into account any of the other forms of evidence as does Richards/Tavtigian – why is that? If you do mean to allow those to be taken into account how would that be done? You claim “Our framework captures the same information currently used to adjudicate variants (ACMG guidelines) and reinterprets it quantitatively and probabilistically in terms of risk and uncertainty.” This does not appear to be the case – I do not see in your framework incorporation of the following criteria from Richards et al: PVS1, PS2, PM1, PM2, PM4, PP1, and PP2.

6. You state “We suggest the proposed framework is useful in cases both where an individual presents with BrS1 or where a variant is discovered incidentally in an individual not presenting with an arrhythmia.” Are you suggesting that what you describe as ‘penetrance’ – the output of your algorithm is the same in diagnostic or screening contexts? I find that difficult to accept.

7. What do you define Brugada syndrome to be? While you roundly criticize Richards et al for being categorical, is it the case that Brugada syndrome is a binary attribute – people either have it or they do not? Given the electrophysiologic nature of this disease, this seems implausible. But even putting that aside I don’t see how you can define a probability of the penetrance of a disorder if you don’t precisely define what it is. This point seems to be consistent with that of one of the peer reviewers.

8. Your assumption that all persons in gnomAD are unaffected is almost certainly wrong. Can you estimate the error introduced by, say 12 people in gnomAD having the disorder? 25 if your estimate of prevalence is off by a factor of 2.0?

9. I do not think that individuals heterozygous for a variant associated with a disorder that is inherited in an autosomal dominant pattern of inheritance should be described as carriers. Better to say ‘heterozygotes’ and reserve ‘carriers’ for disorders inherited in an autosomal recessive pattern.

10. You should use HGVS nomenclature for all variants. Please specify the cDNA change with the first citation of a variant (subsequent cites of that variant can use protein only, but should use proper protein nomenclature and three letter codes e.g., p.(Arg878Cys).)

Reviewer's Responses to Questions

**Comments to the Authors:**

Reviewer #1: The authors built a model to predict Brugada Syndrome (BrS1) penetrance of SCN5A variants and assume that the severity of the variant predicts the probability of having disease symptoms (penetrance). The authors have a good idea to move from categorical classification of variants to a more quantitative measure. However, although the gene function and structure are known to predict variant pathogenicity, it is not clear from this manuscript that they predict penetrance as well. Penetrance is much more complicated than just variant pathogenicity for BrS. They have chosen a disease to study that is somewhat complicated.

Major comments:

1. The authors use data from a collection of 756 publications. The studies recruited participants using different criteria. Given the heterogeneity in the clinical characterization of the clinical phenotyping across studies and the difficulty in making the clinical diagnosis of BrS, I am not confident that this is a reasonable data set upon which to build or test a robust model. Furthermore, because the penetrance for BrS is sex and age dependent in addition to modification with fever and medication, the data upon which to base the models are complicated, and the information on the relevant modifiers is rarely published. The details of the dataset should have been summarized in a table describing the clinical cohort derived from the 756 publications along with a figure of the distribution and frequency of the variants studied.

2. The model design is not clear. To calculate the prior penetrance, one of the features – penetrance density is based on the observed cases and controls. The posterior penetrance is also calculated based on the observed cases and controls. It is unclear if the convergence criterion is meaningful. What does it optimize?

3. How did the authors evaluate the performance of the model? The dataset includes 1000+ variants but only three variants were presented in Figure 1 and 5. The authors used the ClinVar annotation of the three variants to evaluate their estimated penetrance. Pathogenic variants can have penetrance from 0 to 1. The pathogenic variant in the Figure 1 and 5 showed high penetrance, but how do they prove this? Can the authors compare their posterior penetrance with the observed penetrance when the number of carriers is large enough?

4. From Figure 2, the authors seem to be including common variants in their dataset. These common variants have a BrS penetrance of ~3%, which is too high for common benign variants. This implies the prevalence of BrS is high in their cohort which will inflate the penetrance calculation.

5. The prior alpha and beta values depend on the estimated penetrance mean. The authors should not include the benign variants, for which the non-zero penetrance might be due in part to non-genetic reasons or pathogenic variants in other BrS genes.

Reviewer #2: This paper concerns the widely studied medical genetics problem of classifying variants in disease-causing genes with respect to the potential pathogenicity. The authors develop a Bayesian framework to classify variants on a continuous [0,1] range that represents asymptotic (w.r.t. age) penetrance. They apply this framework to hundreds of variants in the gene SCN5A, which is implicated in Brugada syndrome. The manuscript is well written.

I have only two substantial concerns of the which the first is far more important.

Major concerns.

1. I did not see any pointers to available code or to available posteriors in a form that readers can retrieve. To make the method useful to readers who may be interested in applying the method to

other diseases, the authors need to make their code available. To make the Brugada syndrome analysis useful to medical geneticists who study Brugada syndrome, the results (i.e,, prior and posterior distributions) need to be made available.

2. It is unclear how the method handles missing data such as i) variants for which peak current has not been measured or ii) in-frame indels for which several of the sequence analysis tools give no prediction.

Minor concerns.

3. The reasoning by which the feature are equivalent to 19 carriers is unclear. It would help to show explicit calculations that explain why 19 is the right number rather than 18 or 20. Moreover, I do not understand why this number does not depend on the number among the 19 phenotyed carriers who turn out to be affected.

4. How are individuals who carry more than one variant treated in the analysis?

5. How are related individuals carrying the same variant handled in the analysis?

6. How are unaffected young individuals, who may manifest Brugada syndrome at a later not-yet-reached age, handled in the analysis?

7. What is the definition of "match" in the assertion "The penetrance posteriors match classification as presented in ClinVar" in the legend of Figure 1.

8. alpha and beta have not been defined when they are first used at line 156; they are defined later at line 283

9. Line 331 refers to "four commonly used servers" but lists only three methods after the colon.

10. The usage of the definition of distance between residues is not clear, especially for in-frame indels and variants in the promoter and splice sites. As an extreme example, it is unclear why the requirement i not equal to j is included in the first big equation in supplemental methods since there could be two distinct mutations affecting the same amino acid, such as p.P1011L and p.P1011S. Adding some examples of the distance calculation involving different

types of variants and unusual situations would help.

**Have all data underlying the figures and results presented in the manuscript been provided?**

Reviewer #1: No: Need the actual data set they used to try and replicate their findings.

Reviewer #2: No: See major comment 1.

PLOS authors have the option to publish the peer review history of their article (what does this mean?). If published, this will include your full peer review and any attached files.

Reviewer #1: No

Reviewer #2: Yes: Alejandro Schaffer

---

## [Decision Letter · Decision Letter 1]

15 Apr 2020

Dear Dr Kroncke,

Thank you very much for submitting your Research Article entitled 'A Bayesian method to estimate variant-induced disease penetrance' to PLOS Genetics. Your manuscript was fully evaluated at the editorial level and by independent peer reviewers. The reviewers appreciated the attention to an important topic but identified some aspects of the manuscript that should be improved.

We therefore ask you to modify the manuscript according to the review recommendations before we can consider your manuscript for acceptance. Your revisions should address the specific points made by each reviewer.

[LINK]

Yours sincerely,

Leslie Biesecker

Guest Editor

PLOS Genetics

Hua Tang

Section Editor: Natural Variation

PLOS Genetics

This is much improved. One thing I noticed is that the summary of the article needs to be rewritten to shift the narrative to penetrance - it still reads like it is pathogenicity.

Reviewer's Responses to Questions

**Comments to the Authors:**

Reviewer #1: The authors have dramatically improved the manuscript and addressed most of my previous questions. I still do not agree with including common variants. The common/synonymous variants are very likely to be benign, and it is not useful estimate penetrance for predicted benign variants.

The model can estimate penetrance for a previously variant observed in cases. Penetrance for a new variants (not previously observed) cannot be estimated because you cannot form the likelihood probability. If this is true, this should be stated in the limitations.

Reviewer #2: I mostly limited my assessment of the revision to the question: Did the authors address the 10 comments I made on the initial submission.

The authors have not fully addressed my two major comments The authors have fully addressed 6 or 7 of my 8 minor comments. While revising the manuscript

the authors introduced some typos and errors in wording.

Major comment 1.

The authors have not addressed my concern about making the code and data available.

I see some files for this project at

https://github.com/kroncke-lab/resources/tree/master/A%20Bayesian%20method%20to%20estimate%20disease%20penetrance%20from%20genetic%20variant%20properties

However, there is no README file or any other file that I recognize as documentation. The commands

git clone https://github.com/kroncke-lab/resources/tree/master/A%20Bayesian%20method%20to%20estimate%20disease%20penetrance%20from%20genetic%20variant%20properties

git clone https://github.com/kroncke-lab/resources/tree/master

both failed.

Major comment 2.

This may be addressed but I do not understand what is a "missing data pattern" and the corresponding

new text at lines 405-408. Some examples would help.

Minor comment 3.

I understand the response but did not find where this is explained in the text.

Minor comment 4.

OK

Minor comment 5.

OK, except that the newly added text has a typo.

Minor comment 6.

The response in place is OK, but I was surprised to see that the authors claimed in response to the Editor that

""For person X, the variant is either pathogenic or it is not", which misrepresents the challenge of age-dependent

penetrance and this simplistic point of view made it into the revised manuscript at lines 84-85 where the authors wrote

"The pathogenicity of a variant for a specific individual is binary, but unknown".

The authors cannot have it both ways. Either they acknowledge that age-dependent penetrance is

not handled by their method, as suggested by the response to minor comment 6, or they do not.

Minor comment 7.

OK.

Minor comment 8.

OK.

Minor comment 9.

OK.

Minor comment 10.

OK.

Line 19, "to best assess" is a split infinitive

Lines 105 and 125, change "heterozygotic" to "heterozygous"

Line 109, change "which produce" to "that produce"

Line 172, change "used Bland-Altman plot" to "used a Bland-Altman plot"

Line 272, change "59 genes the ACMG recommends" to "59 genes for which the ACMG recommends"

Line 290, change "The result of not accounting for" to "Not accounting for"

Line 373, it is not clear how the indels are scored by the methods such as SIFT; is the score assigned as the worst possible score for any amino acid substitution at the same position?

Line 377, change "For we included" to "We included"

Lines 379-380, change "We do not include intronic variant" to "We did not include intronic variants"

**Have all data underlying the figures and results presented in the manuscript been provided?**

Reviewer #1: Yes

Reviewer #2: No: See my report. I could not download the authors' GitHub repository and it lacks a README.

PLOS authors have the option to publish the peer review history of their article (what does this mean?). If published, this will include your full peer review and any attached files.

Reviewer #1: No

Reviewer #2: Yes: Alejandro A. Schaffer

---

## [Editor Report · Decision Letter 2]

14 May 2020

Dear Dr Kroncke,

We are pleased to inform you that your manuscript entitled "A Bayesian method to estimate variant-induced disease penetrance" has been editorially accepted for publication in PLOS Genetics. Congratulations!

Yours sincerely,

Leslie Biesecker

Guest Editor

PLOS Genetics

Hua Tang

Section Editor: Natural Variation

PLOS Genetics

Comments from the reviewers (if applicable):

Thank you for your responsive revision. I am pleased to recommend to the Senior Editor that your paper be accepted for publication.

**Data Deposition**

http://datadryad.org/submit?journalID=pgenetics&manu=PGENETICS-D-19-01892R2

**Press Queries**

---

## [Editor Report · Acceptance letter]

15 Jun 2020

PGENETICS-D-19-01892R2 

A Bayesian method to estimate variant-induced disease penetrance 

Dear Dr Kroncke, 

We are pleased to inform you that your manuscript entitled "A Bayesian method to estimate variant-induced disease penetrance" has been formally accepted for publication in PLOS Genetics! Your manuscript is now with our production department and you will be notified of the publication date in due course.

With kind regards,

Matt Lyles

PLOS Genetics

On behalf of:
